# Canal Transportation and Volumetric Dentin Removal Abilities of Ni-Ti Rotary File Systems in Curved Primary Root Canals: CBCT Study

**Aylin İslam [1,2,*], Gürkan Ünsal [2,3] and Alaa Almashharawi [1]**

[1] Department of Pediatric Dentistry, Faculty of Dentistry, Near East University, Nicosia 99138, Cyprus; alaa.almashharawi@neu.edu.tr

[2] Desam Institute, Near East University, Nicosia 99138, Cyprus; gurkan.unsal@neu.edu.tr

[3] Department of Dentomaxillofacial Radiology, Faculty of Dentistry, Near East University, Nicosia 99138, Cyprus

[*] Correspondence: aylin.islam@neu.edu.tr; Tel.: +90-(392)-680-20-30; Fax: +90-(392)-680-20-25

**Abstract:** Current improvements in nickel-titanium (Ni-Ti) rotary file systems have created a paradigm shift in the root canal therapy of primary teeth. Therefore, it is necessary to perform a comprehensive evaluation regarding the efficiencies of newly manufactured instruments for different parameters. The current study was conducted to evaluate the abilities of RaceEvo, R-Motion, ProTaper Gold (PTG) systems in curved primary root canals with regard to the patterns of canal transportation and volumetric dentin removal by using cone-beam computed tomography (CBCT). Two experimental sets were designed following the determination of experimental groups by using pre- and post-operative CBCT data: canal transportation and volumetric dentin removal. The highest amount of canal transportation was significantly detected in the PTG group in comparison to RaceEvo and R-Motion groups. When the mean values of volumetric dentin removal data were analyzed across all groups, the PTG group again exhibited the significantly highest value of dentin removal volumetrically, compared to RaceEvo, R-Motion and manual instrumentation groups. It is possible to state that R-Motion and RaceEvo rotary systems could be used as reliable alternatives without causing adverse mechanical effects and maintaining the original root canal anatomy of curved primary root canal systems compared with PTG rotary systems and manual instrumentation, with a high diagnostic sensitivity of CBCT in pediatric endodontics when the alternative methods are not adequate.

**Keywords:** cone-beam computed tomography; nickel-titanium rotary files; canal transportation; volumetric dentin removal; RaceEvo; R-motion; ProTaper Gold

## 1. Introduction

Today, endodontic approaches (pulpectomy procedure) could be considered as the most appropriate solution to provide maintenance of primary molars diagnosed with irreversible pulpitis or pulpal necrosis in the oral cavity until their required physiological shedding time [1,2]. In correlation with permanent dentition, the clinical success rate of root canal treatment in primary dentition is mainly related to perfect biomechanical preparation, the removal of microorganisms, and infected pulpal remnants from the root canal system [3]. However, complexities or variations in root canal anatomy and the high root resorption probability of primary teeth jeopardize and complicate root canal therapy (RCT) [4]. The characteristics of primary molars with curved and shorter roots, ribbon-shaped morphology, and ectopic surface resorption, especially, must be taken into consideration during RCT [5]. Along with these challenging characteristics, the selection of a suitable instrument that can preserve the original shape of the root canal without creating any variations and providing regular dentin removal should also be kept in mind

during biomechanical preparation. Generally, manual instrumentation is still preferred as the main option for the preparation of the primary root canals. However, various clinical complications during manual instrumentation such as perforations, ledge formation, canal transportation, dentinal cracks, dentin compaction, and broken files were reported [6,7]. Firstly, the rotary file systems for endodontic treatment approaches were introduced in primary dentition by Barr and colleagues in 2000 to overcome these complex anatomical morphology and clinical errors [8]. The capability of rotary systems that for slight debris extrusion through the apical foramen to the surrounding periapical tissues is considered one of the main advantages. Furthermore, significantly minimized chair time for pediatric dental patients, no excessive dentinal tissue removal, and no canal transportation in curved canals were reported as the other major advantages of rotary systems [9,10]. Over the last few decades, numerous new generations of rotary files systems were developed. Specifically, the clinical efficiencies (less canal transportation, successful in preserving root dentinal walls, retention of original root canal shape) of ProTaper Gold, ProTaper Next, Wave One, and Kedo-S files in primary and permanent dentition were reported by many authors [11–14].

Recently, the ProTaper Gold (PTG) rotary file system, which works with the principle of a rotating mechanism, was produced. PTG has a progressive taper that results in a more effective cutting ability and improved safety on dentinal tissue [15]. On the other hand, reciprocating systems were developed as an alternative to the continuous rotation for the preparation of curved canals by using a single file. In this technique, the safety of root canals is achieved by decreasing the possibility of fractures on root canal walls and removing less dentinal tissue [6]. Most recently, RaceEvo and R-motion rotary systems were developed with greater flexibility, cutting efficiency and improved cross-sectional design that are significant factors in the canal transportation of curved root canals. However, to the best of our knowledge, there are no published data available concerning the three-dimensional (3D, e.g., volumetric) and two-dimensional (2D, e.g., canal transportation) efficiencies of these newly designed rotary systems on primary curved root canals. Thus, from a clinically significant point of view, it is imperative to assess the relevant parameters, shaping quality regarding canal transportation, canal straightening, and unnecessary hard dental tissue removal conditions during the preparation of curved primary root canals to perform successful long-term clinical trials with minimally invasive dentistry.

The present study was conducted to comparatively evaluate the canal transportation and determine the volumetric analysis of dentin removal patterns in curved primary root canals after instrumentation with Hand K-file, ProTaper Gold, RaceEvo, and R-Motion in three dimensions by using cone-beam computed tomography (CBCT).

## 2. Materials and Methods

### 2.1. Ethical Approval

Experimental steps of the present experimental in vitro study were performed in the clinics of Pediatric Dentistry and the Dental and Maxillofacial Radiology Departments at Near East University. The study was conducted according to the guidelines of the Declaration of Helsinki, and approved by the Ethics Committee of Near East University (NEU/2021/87-1253). All subjects gave their written informed consent for inclusion before they participated in the study.

### 2.2. Selection and Preparation of Specimens

The inclusive criteria were determined based on external/internal root resorption, calcifications, previous endodontic treatments, fracture/crack lines, and root canals with two-thirds of the root length remaining. During the detailed selection of specimens, preoperative diagnostic images were taken by CBCT to eliminate the root canals with calcification and internal root resorption. Specimens were determined according to their similarities in terms of root canal lengths and the root canal curvatures (root canals with moderate curvatures were included: 10–20°) were based on Schneider's technique [16].

The speed images of excluded specimens from the current study are shown in Figure 1. In total, sixty (n = 60) curved primary molar root canals were selected according to the predetermined inclusive criteria.

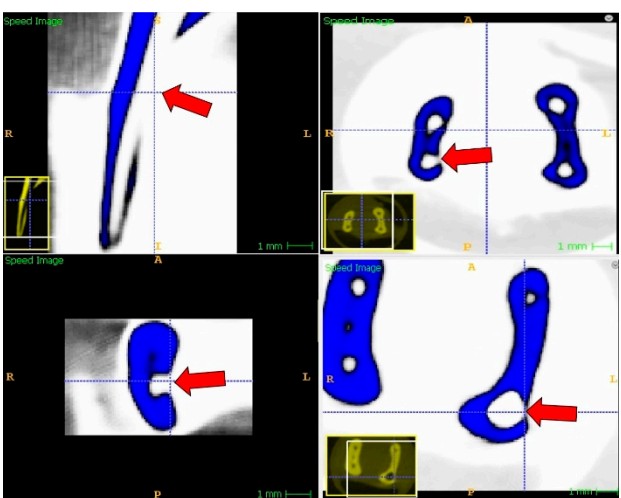

**Figure 1.** Speed images of perforated root canals that were excluded from the study.

Initially, the removal of coronal caries was performed by using a diamond round bur (Meisinger, Germany). Then, access cavities were opened, and pulp chambers were irrigated with 1.5% sodium hypochlorite (NaOCl, Cerkamed, Poland) prior to determining the working length of each root canal. The working length was determined by using a size 10 K-file (VDW, Munich, Germany), which was manually inserted into the root canals until the visual apical foramen and 1 mm short value from this initial measurement wasrecorded as the working length for mechanical root canal preparation.

### 2.3. Preoperative CBCT Investigation of Specimens

The selected specimens were examined by CBCT preoperatively before the mechanical root canal preparation procedure after each specimen was placed into one wax box. Imaging parameters for CBCT were 85 kVp, 6 mA, 14.4 s with a 55 mm × 50 mm field of view (FOV) and at a 100 μm isotropic voxel size.

### 2.4. Determination of Groups

The selected specimens were randomly divided into four groups (n = 15/each group) regarding the type of instrumentation file.

In Group I (Manual Instrumentation), a total of 15 root canals were manually instrumented using hand nickel-titanium (Ni-Ti) K-files (VDW, Munich, Germany) in accordance with the step-back technique. Each consecutive file was used with the same trend until the apical preparation size reached a 30 Ni-Ti K-file. Repetition with the 30 Ni-Ti K-file was performed to prevent ledge formation.

In Group II (ProTaper Gold), a total of 15 root canals were shaped with the ProTaper Gold Rotary System until they reached F3 Size (Maillefer, Dentsply, Switzerland). The parameters of the X-smart Plus motor (Maillefer, Dentsply, Switzerland) were set as 300 rpm and a torque of 5 Ncm. First, Sx was used with a brushing action, until it encountered resistance. Following Sx, S1 was used again with the brushing action mode, until 1 mm short of the working length, to ascertain radicular access. Afterward, S2 was used in the same manner until it reached the working length. The finishing preparation of root canals was instrumented with F1, F2, and F3 finishing files in a non-brushing action, with each insertion deeper than the previous insertion until the working length was reached.

In Group III (RaceEvo), a total of 15 root canals were prepared using the RaceEvo Rotary System (FKG, Switzerland), consisting of a size 15/RE1, size 25/RE2 and a size

30/RE3 with 0.04 tapers. The parameters of the X-smart Plus motor (Maillefer, Dentsply, Switzerland) were adjusted to the recommended speed range at 800 rpm with continuous rotation and a torque of 1.5 Ncm. Initially, the RE1 file was used until the working length was reached to perform a glide path. Then, the RE2 file was used, and the final instrumentation was performed using the RE3 file until the definitive working length. Once each instrument reached the working length, it was immediately removed to prevent over-enlargement of the apical foramen.

In Group IV(R-Motion), a total of 15 root canals were shaped with the R-Motion Rotary System (FKG, Switzerland), including a size 15/R-Motion Glider (0.03 taper), a size 25/R-Motion 25 (0.06 taper) and a size 30/R-Motion 30 (0.04 taper). The parameters of the X-smart Plus motor (Maillefer, Dentsply, Switzerland) were arranged according to recommendations which used only the standard reciprocating mode (170° counterclockwise and 50° clockwise rotation modes). First, the mechanized R-Motion Glider instrument was used to establish a glide path until it reached working length. Afterward, the shaping procedure of root canals was performed with R-Motion 25 and R-Motion 30 files. Again, once each instrument reached the working length, it was immediately removed to prevent over-enlargement of the apical foramen.

Throughout the preparation, the root canals were regularly and frequently irrigated with 1.5% NaOCl and saline solution after each instrument had been used.

### 2.5. Postoperative CBCT Investigation of Specimens and the Determination of Canal Transportation

Following the preparation of each root canal in each experimental group, postoperative CBCT images were taken, using the parameters of 85 kVp, 6 mA, and 14.4 s with a 55 mm × 50 mm FOV. Then, the determination of the canal transportation step was designed and measured in a three-dimensional (3D) environment by drawing an illusory line located above the furcation area for each prepared root canal. All root canals had different working lengths; therefore, the evaluation of each root canal was made individually by dividing the length of the root canal into the superior 1/3, middle 1/3, and apical 1/3. To avoid any error which may have been caused by minor angulation differences, pre-operative and post-operative axial slices of each root canal were superimposed using On Demand 3Ds, On Demand3D Fusion (Cybermed Inc., Seoul, Korea) software. This process was performed on pre-operative and post-operative CBCT images to perform an overall evaluation of canal transportation for each root canal. Following the implementation of both designs, the differences between pre-operative and post-operative calculations were determined as the overall canal transportation values.

### 2.6. Volumetric Determination of Dentin Removal

Pre-operative CBCT images (T1) and post-operative CBCT images (T2) were taken using an Orthophos SL 3D CBCT Unit (Dentsply Sirona, Erlangen, Germany), and the images were stored as DICOM files. All the DICOM files were analyzed with the ITK-SNAP version 3.6.0 (www.itksnap.org) open-access image analysis software, which was used to perform volumetric analysis on the 3D images. Active Contour (Snake) Segmentation Mode was used to eliminate the air, crown, and other roots outside of the region of interest (ROI) and to have faster segmentation. "Thresholding" was used for the pre-segmentation mode and the threshold for the segmentation of root canals was set between "−1024" and "1200" (Figure 2). Following the segmentation, bubbles with a 0.4 mm radius were placed inside the "white" area, which represents the multiple root canal spots, and contour evolution was executed. The "Volumes and statistics" option was checked in the segmentation toolbox and the voxel count, volume ($mm^3$) and intensity mean with standard deviation were noted. Volumes were recorded for both T1 (VT1) and T2 (VT2) images, and volumetric dentin removal was calculated as VT1-VT2. This method was used for every single root canal which met the inclusion criteria. The multiplanar aspects of a single root canal are shown in Figure 3.

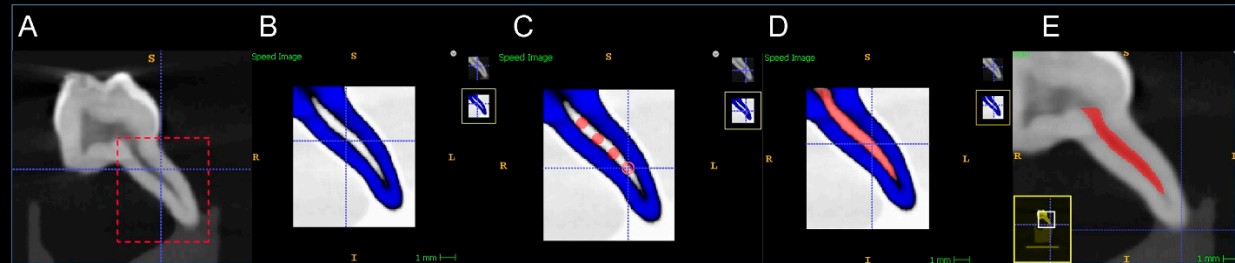

**Figure 2.** Thresholding for segmentation of the primary root canal. (**A**) Active contour "Snake" segmentation window; (**B**) Pre-segmentation speed image of a single primary root canal with threshold between −1024 and 1200 GV; (**C**) Bubble placement to initialize the contour with 1.20 mm radius red bubbles; (**D**) Post-execution speed image of the root canal (red areas); (**E**) Final image of the primary root canal.

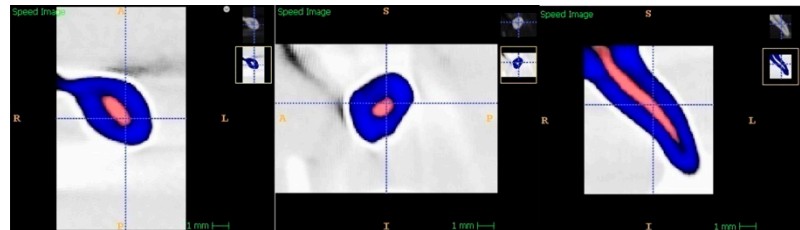

**Figure 3.** Multiplanar images of a single primary root canal.

### 2.7. Statistical Analysis

Initially, obtained data for each analysis were checked for normality. Not all data were normally distributed; therefore, non-parametric Kruskal–Wallis and Dunn's post hoc tests were used for the analysis of the differences between the comparable data groups. GraphPad Prism software (version 8.1.1, San Diego, CA, USA) and SPSS (version 20.0, Chicago, IL, USA) were utilized for analyses. The significance level was accepted at 0.05 for all statistical analyses.

### 3. Results

The pre- and post-measurements on the CBCT images of the present study in the experimental groups were compared and evaluated for two major determinations, consisting of canal transportation and volumetric dentin removal.

### 3.1. Assessment of Canal Transportation

Significant differences were observed in the multiple comparisons of canal transportation data between the mean values of each group. The highest degree of canal transportation was significantly detected in the ProTaper Gold rotary system ($0.2255 \pm 0.015$). These data were found to be statistically significant in comparison with RaceEvo and R-Motion rotary groups ($0.1329 \pm 0.013/p = 0.0002$; $0.1386 \pm 0.014/p = 0.0006$, respectively), whereas no statistical differences were observed among all other comparisons regarding canal transportation data. The mean values of overall canal transportation measurements in all experimental groups are presented in Table 1 and Figure 4. Additionally, the comparative measurements (preoperative–postoperative) of root canal transportation are represented in Figure 5.

### 3.2. Assessment of Volumetric Dentin Removal

When the mean values of volumetric dentin removal data were analyzed across all groups, the ProTaper Gold rotary system ($2.015 \pm 0.074$) again exhibited the significantly highest value of dentin removal as compared, volumetrically, to the RaceEvo, R-Motion and Manual instrumentation groups ($0.4629 \pm 0.018$; $0.4957 \pm 0.018$; $0.8427 \pm 0.12$, respectively/$p = 0.000$ for all comparisons), whereas no statistical difference was observed in the

volumetric dentin removal comparison between RaceEvo and R-Motion groups ($p > 0.05$). Moreover, the higher volumetric dentin removal value was demonstrated in manual instrumentation group when compared with RaceEvo and R-Motion rotary systems ($p = 0.014$; $p = 0.027$, respectively). The mean values of volumetric dentin removal measurements in all experimental groups are given in Table 2 and Figure 6. Additionally, three-dimensional preoperative and postoperative reconstruction images are illustrated in Figure 7.

**Table 1.** Multiple comparisons of canal transportation across experimental groups. (Results are expressed as mean values $\pm$ SEM). (*** $p < 0.001$).

| Multiple Comparisons of Canal Transportation across Experimental Groups | | | *p*-Value |
|---|---|---|---|
| Race Evo vs. R-Motion | $0.1329 \pm 0.013$ | $0.1386 \pm 0.014$ | >0.9999 |
| Race Evo vs. ProTaper Gold | $0.1329 \pm 0.013$ | $0.2255 \pm 0.015$ | 0.0002 *** |
| Race Evo vs. Manual | $0.1329 \pm 0.013$ | $0.1781 \pm 0.016$ | 0.2086 |
| R-Motion vs. ProTaper Gold | $0.1386 \pm 0.014$ | $0.2255 \pm 0.015$ | 0.0006 *** |
| R-Motion vs. Manual | $0.1386 \pm 0.014$ | $0.1781 \pm 0.016$ | 0.4603 |
| ProTaper Gold vs. Manual | $0.2255 \pm 0.015$ | $0.1781 \pm 0.016$ | 0.2126 |

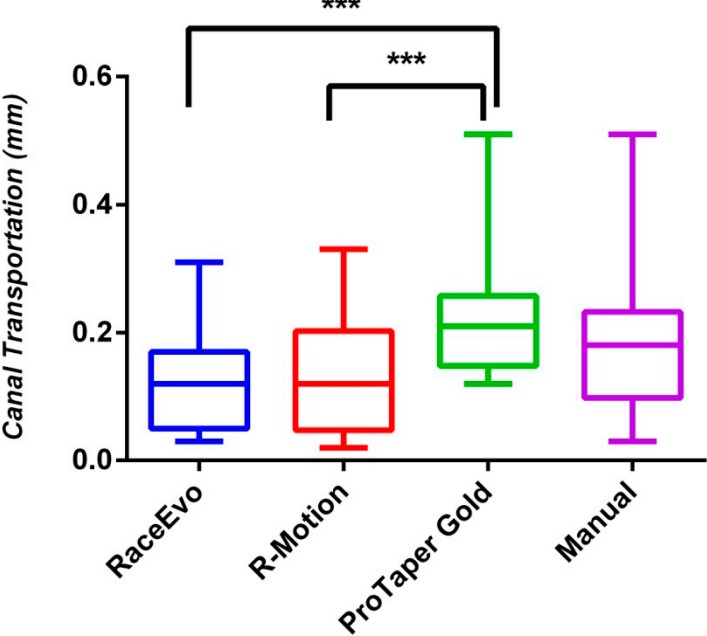

**Figure 4.** Canal transportation across experimental groups. Results are expressed as mean values $\pm$ SEM. (*** $p < 0.001$).

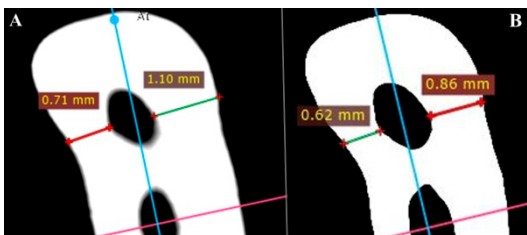

**Figure 5.** Measurements of root canal transportation (**A**) before instrumentation and (**B**) after instrumentation.

**Table 2.** Multiple comparisons of volumetric dentin removal across experimental groups. (Results are given as mean values $\pm$ SEM). (*** $p < 0.001$, * $p < 0.1$).

| Multiple Comparisons of Volumetric Dentin Removal across Experimental Groups | | | *p*-Value |
|---|---|---|---|
| Race Evo vs. R-Motion | $0.4629 \pm 0.018$ | $0.4957 \pm 0.017$ | 0.679 |
| Race Evo vs. ProTaper Gold | $0.4629 \pm 0.019$ | $2.015 \pm 0.074$ | 0.000 *** |
| Race Evo vs. Manual | $0.4629 \pm 0.020$ | $0.8427 \pm 0.11$ | 0.014 * |
| R-Motion vs. ProTaper Gold | $0.4957 \pm 0.017$ | $2.015 \pm 0.074$ | 0.000 *** |
| R-Motion vs. Manual | $0.4957 \pm 0.018$ | $0.8427 \pm 0.11$ | 0.027 * |
| ProTaper Gold vs. Manual | $2.015 \pm 0.074$ | $0.8427 \pm 0.12$ | 0.000 *** |

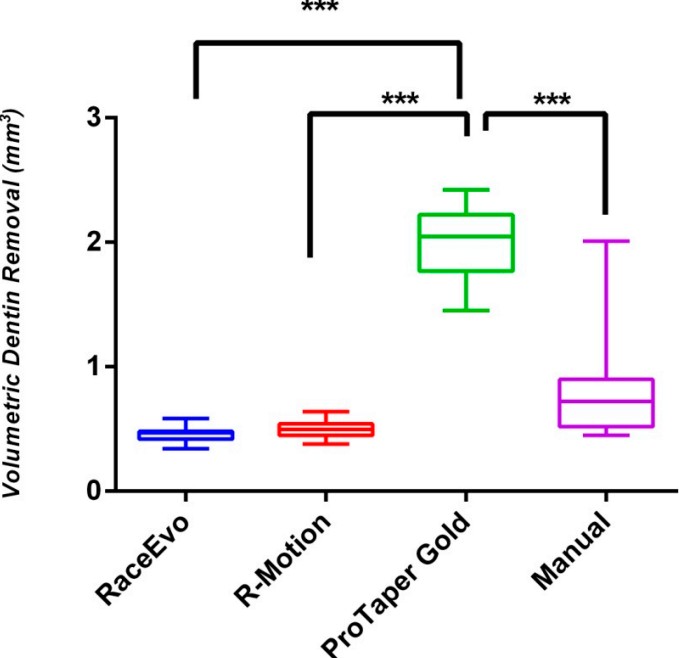

**Figure 6.** Volumetric dentin removal across experimental groups. Results are given as mean values $\pm$ SEM. (*** $p < 0.001$).

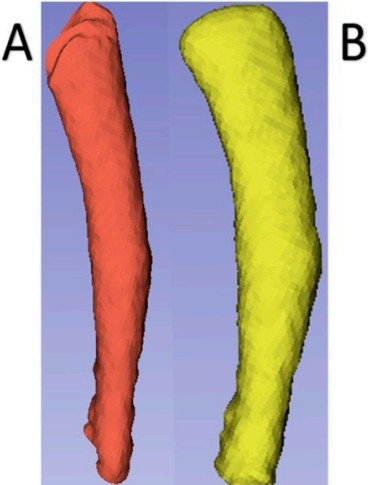

**Figure 7.** Buccolingual views of preoperative (red) and postoperative (yellow) root canal three-dimensional (3D) reconstruction images of a primary mandibular molar tooth (reconstructed from an exported surface mesh STL data). (**A**) Preoperative-3D buccolingual view of a root canal; (**B**) Postoperative-3D buccolingual view of a root canal.

In the present study, the radiographic success of all the tested manual and rotary instrumentation systems was also evaluated. There were four (4) primary root canals were detected with perforations in the ProTaper Gold (PTG) rotary file system and also one (1) case with perforation was observed in the manual instrumentation group while no cement perforation was present in both RaceEvo and R-Motion group after preparation procedure among all tested groups. (Perforated root canals were replaced with new primary root canals and the procedures were repeated).

## 4. Discussion

In addition to the effective enlargement of a root canal by providing an opportunity for the removal of organic and inorganic debris, the accomplishment and preservation of an original continuous/tapered root canal anatomy from apical to coronal through adequate dentin thickness during the instrumentation of tortuous root canals of primary molars are also noteworthy [17]. However, a number of procedural errors, such as canal transportation and inappropriate dentin removal complications, can lead to subsequently decreased fracture resistance, endangering original canal curvature, increasing the risk of debris extrusion and postoperative pain [18,19]. Current improvements in rotary Ni-Ti file systems create a paradigm shift in the root canal therapy of primary teeth. Therefore, it is necessary to perform a comprehensive evaluation regarding efficiencies of newly manufactured instruments for root canal shaping in terms of different parameters. Hence, the current study was conducted to compare and evaluate the abilities of relatively novel RaceEvo, R-Motion, ProTaper Gold systems in curved primary root canals with regard to the patterns of canal transportation and volumetric dentin removal by using CBCT.

Numerous methodologies involving serial sectioning, microscopic analyses, muffle systems, radiographs, silicone impressions, endodontic tubes, multislice computed tomography, CBCT, etc., were described to assess different instrumentation parameters [20]. CBCT has become more popular and is a favorable alternative tool in the contemporary endodontic field, especially for the evaluation of three-dimensional (3D) root canal anatomy, determination of the amount of dentin for removal, measurements of canal transportation, centering ability, and conducting comparisons between before and after the instrumentation of root canals. Additionally, CBCT can be used to measure and calculate the volumetric units of enamel, dentin, and pulp tissues [20,21]. In the current study, CBCT was preferred to determine the inclusive criteria and compare the parameters of overall canal transportation and volumetric dentin removal patterns in curved primary root canals prepared by manual and rotary instrumentation systems.

The determination of canal transportation patterns was selected as a primary parameter to assess the shaping effectiveness of tested instruments in each experimental group. The canal transportation was evaluated overall for each prepared root canal length. For overall canal transportation analysis among all groups, the results showed that the newly generated RaceEvo and R-motion rotary systems clearly exhibited fewer canal transportation complications compared to the ProTaper Gold (PTG) rotary system, while preserving the original curved root canal shapes following instrumentation. The highest mean transportation values were observed in the PTG group. These important differences could be attributed to the manufactured techniques and characterizations of the tested instruments. The instruments of R-Motion rotary systems were designed with thinner core sizes, increased flexibility, rounded triangular cross-sections with sharp cutting edges and new, optimized file tips [22]. With that in mind, concerning these developed properties, the R-Motion rotary systems provide better respect to the curved primary root canal anatomies without creating significant canal transportation complications while preserving and reducing the stress on dentinal tissue compared to PTG systems.

Moreover, the difference in mean transportation levels between R-Motion and PTG rotary systems could be elucidated by the lower screwing effect design of R-Motion rotary systems that allow the clinician to provide a higher control efficiency during the progression in root canals [22]. A similar efficiency in canal transportation with R-Motion was also

observed in the RaceEvo rotary system compared to PTG and manual instrumentation. RaceEvo rotary systems are engineered by a heat-treated process and created with a higher rotation speed performance design. The combination of the sharp-edged, triangular design of RaceEvo with high-speed performance and heat treatment process delivers numerous advantages, such as higher flexibility, greater resistance against cyclic fatigue, and improved cutting efficiency [23]. Thus, the outstanding performance of RaceEvo in preserving the original root canal shape without creating adverse canal transportation effects could be clarified from the present results.

Although the results of the present investigation could not be compared directly with those of Hashem et al. [24] and Agarwal et al. [25] due to the usage of different systems and methodologies, their prominent results concerning PTG systems were consistent with the results of the current study. In both of these studies, it was reported that PTG systems exhibited the higher amount of canal transportation rates. Instrument tapering is another point that has an inverse effect on canal transportation patterns. Studies related to instrument tapering and canal transportation relationships have revealed that Ni-Ti files with taper sizes greater than 0.04 (4%) are not recommended due to their apical enlargement and higher canal transportation ratios in curved root canals [26,27]. In the present study, both RaceEvo and R-Motion systems were used with 4% tapers, whereas the PTG system was operated until F3 (0.09/9% taper) size to provide standardization. Therefore, the detection of a higher canal transportation rate in the PTG system in comparison to RaceEvo and R-Motion systems might be better explained and is consistent with previous studies [24,27,28].

Studies on the efficiency of these newly designed R-Motion and RaceEvo rotary systems on primary curved root canals are very scarce. Therefore, the possibility of an exact comparison of the tested newly designed systems regarding patterns of canal transportation and volumetric dentin removal with similar studies is very limited.

Importantly, the quantitative volumetric dentin removal patterns of used instruments in curved primary root canals were also comparatively analyzed in the current study to better understand the usability of these new rotary instruments in the primary dentition. The results of volumetric dentin removal analysis showed significant differences among the groups. In the present study, the highest mean value of volumetric dentin removal was observed in the PTG rotary group. Generally, the PTG rotary and manual instrumentation groups exhibited higher mean values of volumetric dentin removal when compared to RaceEvo and R-Motion rotary groups. This aggressive attitude of the PTG rotary group in curved primary root canals may be clarified by the convex triangular cross-section and progressive taper design that enhances the cutting action and leads to aggressive cutting side effects on the dentin surface [29]. Even though there was little difference in the manual instrumentation group compared to the RaceEvo and R-Motion rotary groups regarding the pattern of volumetric dentin removal, the RaceEvo and R-Motion rotary groups produced superior results. This situation could be attributed to the intensive performance of the operator during instrumentation. There was no quantitative volumetric analysis performed; however, the results of Kummer et al. [30] and Musale et al.'s [20] studies are partially consistent with the results of the present study, in that the amount of volumetric dentin removal was significantly higher with the manual instrumentation technique compared with rotary groups, except for the PTG group.

In recent years, Ni-Ti wires have gained popularity in the dentistry field due to their shape memory and super elasticity abilities. These properties of the alloys are gained by a specific phase transformation thatoccurs as a result of the transition from austenite to martensite forms [31]. The tested RaceEvo and R-Motion rotary groups in the current study comprised medical-grade Ni-Ti alloy with a triangular symmetrical cross-section with sharp edges. The active parts of both rotary groups underwent exclusive proprietary heat treatment which triggered the phase transition mentioned above (between martensite and austenite), just below body temperature, i.e., between 32 °C and 35 °C [22,23]. Additionally, it was reported that the thermomechanical treatment of Ni-Ti alloy instruments does not allow the straightening of instruments in curved root canals during preparation and

provides less root canal transportation with acceptable shaping abilities [32]. On the other hand, a patented heat-treatment mechanism called controlled memory wire (CM wire) is often used in PTG rotary systems instead of heat-treated Ni-Ti wire, distinct from other all tested rotary and hand instruments in the study. It was proven that instruments produced using CM wire exhibit higher flexibility and greater resistance during root canal preparation by reducing the risk of some complications, such as deviation, instrument fracture, and perforation [33,34]. Despite these improved physical characteristics of PTG rotary systems with CM wire manufacturing when interpreting the tested instruments from the viewpoint of influence of shape and alloy features on canal transportation and volumetric dentin removal parameters, RaceEvo and R-Motion rotary groups exhibited superior abilities in terms of canal transportation and volumetric dentin removal parameters than the PTG group and manual instrumentation. These unexpected differences could be explained by the progressive taper (9% taper size—F3) and enhanced cutting efficiency with a convex triangular cross-sectional design [29].

Another important issue that should be discussed and compared is *"the importance and effect of reciprocating motion"*. The reciprocating motion was first introduced by Roane et al. [35] and theorized as a balanced force during instrumentation on the basis of laws of action and reaction. It is important to emphasize that reciprocating motion minimizes the flexural stresses and provides an opportunity to increase the canal centering ability by reducing the taper lock effect of the instrument during canal preparation. From the viewpoint of pediatric endodontics, the preference of reciprocating motion instead of continuous rotation could reduce the stress and required time for preparation in curved root canals [36]. With these in mind, the obtained superior results of R-Motion regarding the patterns of canal transportation and volumetric dentin removal compared to PTG and manual instrumentation groups may be supported and clarified from a differential aspect. Additionally, the superiority of reciprocating motion in the present study was partially consistent with the results of Prabhakar et al. [37].

A major limitation of the current study was the Hounsfield units ($HU_S$), which were used for thresholding, and are not as reliable as the CT units. Various issues related to limited-field CBCT geometry, basic principles of radiation physics, and the limitations of currently used reconstruction algorithms are associated with the use of HUs in CBCT. However, it is also known that CT units do not have isotropic voxels and small slice thicknesses such as in CBCT units (<1 mm), which are useful for examining small, narrow anatomical structures such as root canals. HUs are considered reliable at these scales. Occasionally, grayscale values (GVs) are used in CBCT instead of HUs, and some inaccuracies occur, such as [38–41]:

- The presence of beam hardening, cupping, and doming artifacts;
- Divergence of the X-ray beam;
- Axial slice variability due to different masses of each slice;
- High image noise (which has a minor effect on small-region examinations such as in the current study);
- Absence of GV calibrations in CBCT units by some manufacturers;
- Differences in GVs for the same matter in different CBCT models.

On the other hand, Pauwels et al. reported that GVs may differ depending on the CBCT device, amount of the mass inside and outside of the field of view, central/peripheral positioning of the matter, and exposure parameters [38,40]. The main aim ofthis study was to compare different root canal instrumentation systems; therefore, the same CBCT unit with the same thresholding, imaging parameters and positioning for all of the images was used in order to reduce the effects of this limitation.

## 5. Conclusions

Within the abovementioned restrictive points and among the tested parameters of the current study, it is possible to state the following conclusions:

- The newly generated R-Motion and RaceEvo rotary systems exhibited superior preparative results by producing less overall canal transportation and less volumetric dentin removal compared to ProTaper Gold and manual instrumentation techniques;
- R-Motion and RaceEvo rotary systems could be used as a reliable alternative without causing adverse mechanical effects and maintaining original root canal anatomy of curved primary root canal systems compared to ProTaper Gold rotary systems with manual instrumentation;
- Generally, more supportive studies evaluating the abilities of newly introduced rotary systems on curved primary root canals using detailed CBCT analyses are required.

Moreover, further investigations determining the influence of more severe and complex primary root canal curvatures, various strict environmental conditions, or different surface treatment techniques on newly designed instruments with further in vivo studies are required and certainly recommended.

**Author Contributions:** Conceptualization, A.İ., G.Ü. and A.A.; Methodology, A.İ. and G.Ü.; Software, A.İ. and G.Ü.; Validation, A.İ. and G.Ü.; Formal Analysis, A.İ. and G.Ü.; Investigation, A.İ., G.Ü. and A.A.; Resources, A.İ., G.Ü. and A.A.; Data Curation, A.İ. and G.Ü.; Writing—Original Draft Preparation, A.İ., G.Ü. and A.A.; Writing—Review and Editing, A.İ. and G.Ü.; Visualization, A.İ. and G.Ü.; Supervision, A.İ.; Project Administration, A.İ.; Funding Acquisition, A.İ., G.Ü. and A.A. All authors have read and agreed to the published version of the manuscript.

**Funding:** This research received no external funding.

**Institutional Review Board Statement:** The study was conducted according to the guidelines of the Declaration of Helsinki, and approved by the Institutional Review Board (or Ethics Committee) of Near East University ((NEU/2021/87-1253, 28 January 2021).

**Informed Consent Statement:** Written informed consent was obtained from the patients to publish this paper.

**Conflicts of Interest:** The authors declare no conflict of interest.

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
