# Peer review of "Canal Transportation and Volumetric Dentin Removal Abilities of Ni-Ti Rotary File Systems in Curved Primary Root Canals: CBCT Study"

_applsci, doi:10.3390/app11199053_

Round 1

Reviewer 1 Report

The study establishes a comparison between the different systems of rotary nickel-titanium files in the primary dentition using the CBCT as an imaging technique to perform the measurements. The items analyzed are canal transportation and volumetric dentin removal.

I have doubts about the justification for performing two CBCTs in children with the radiation that this implies. I understand that the volumetric calculation requires 3D images, but a pre and post are necessary for the study but not so much for the diagnosis and treatment of the patient.Regarding the planning of the trial, it is elaborate and takes into account all possible details that could modify the results, including the angulation of the canals.Although it is a little explored field, I recommend updating the bibliography since in this year 2021 there are quite interesting publications on the subject.Another factor that would have been interesting to assess is the success rate of the procedure in relation to the different factors taken into account. I miss seeing this reflected as a justification for carrying out the study.

Author Response

Dear Dr. Reviewer,

First of all, we are very grateful to you to for your suggestions to improve our manuscript. We are pleased to answer your questions and make revisions according to your suggestions.

Yes

Can beimproved

Must be improved

Not applicable

Does the introduction provide sufficient background and include all relevant references?

( )

( )

(x)

( )

Is the research design appropriate?

( )

(x)

( )

( )

Are the methods adequately described?

(x)

( )

( )

( )

Are the results clearly presented?

( )

(x)

( )

( )

Are the conclusions supported by the results?

( )

(x)

( )

( )

Comments and Suggestions for Authors

The study establishes a comparison between the different systems of rotary nickel-titanium files in the primary dentition using the CBCT as an imaging technique to perform the measurements. The items analyzed are canal transportation and volumetric dentin removal.

I have doubts about the justification for performing two CBCTs in children with the radiation that this implies. I understand that the volumetric calculation requires 3D images, but a pre and post are necessary for the study but not so much for the diagnosis and treatment of the patient.Regarding the planning of the trial, it is elaborate and takes into account all possible details that could modify the results, including the angulation of the canals.

A: As authors we appreciate for your valuable comment. We understand your point and concern of radiation dose in children. However, our motivation was not to encourage excessive CBCT usage in children, the main aim of our research was  to demonstrate the significant difference of dentin removal in different rotary systems. In line with our purpose, pre and post CBCT imagings were used to assess the shaping abilities in terms of canal transportation and volumetric dentin removal in different rotary systems prior to performing clinical trials. Hence, it could be possible to establish an optimal - clinical alternative approach and also select the best rotary system regarding shaping ability (non-invasive) for primary root canal treatments.

Micro-CT could also had been used for this kind of study but since it is not applicable for the clinic yet, we would never be sure if the difference between the rotary systems would be observable with dental imaging systems. In other words, we were not trying to promote pre-root canal treatment and post-root canal treatment CBCT imaging but we were trying to demonstrate dentin removal with an imaging which has small field of view, isotropic and small voxels.

Although it is a little explored field, I recommend updating the bibliography since in this year 2021 there are quite interesting publications on the subject.

A: Thank you for your valuable comment. Whole bibliography was checked and up-dated. Specifically, the references before 2010 year were tried to remove and changed with up-dated interesting publications related with our subject. The up-dated references were highlighted as red color in the Reference section.

Another factor that would have been interesting to assess is the success rate of the procedure in relation to the different factors taken into account. I miss seeing this reflected as a justification for carrying out the study.

A: We would like to thank you for your constructive comment. We tried to give more details in the Results section regarding radiographic success of the all tested manual and rotary systems. Our added details regarding your comment are as indicated below.

The sentence ‘‘In the present study the radiographic success of the all tested manual and rotary instrumentation systems were also evaluated. There were four (4) primary root canals were detected with perforations in the ProTaper Gold (PTG) rotary file system and also one (1) case with perforation was observed in the manual instrumentation group while no cement perforation was present in both RaceEvo and R-Motion group after preparation procedure among all tested groups. [Perforated root canals were replaced with new primary root canals and the procedures were repeated] ’’.

***Also, the speed image of the perforated root canals that were excluded from the study was represented in Figure 1.  

Reviewer 2 Report

The article presents an interesting study. However, I have some comments:

Title

I suggest changing the title to:

Canal Transportation and Volumetric Dentin Removal Abilities of Ni-Ti Rotary File Systems in Curved Primary Root Canals: CBCT Study

Introduction

Please summarize first 10 lines into 1-2 sentences.

Results

Please divide results into subsections.

Please explain in method section what do you mean with “Mean1’ and ‘Mean2’ (table 1).

Can you present table 1 as box and whiskers plot ?

Figure 4: was there any statistically significant difference between PTG and manual group? The information is missing on the graph.

Please place capitation of the table above the table.

Discussion

Lines 271-316 please divide this paragraph into smaller sections.

Conclusions

This conclusion No 3 do not answer the aim and is not supported by results.

References

Pleas up-date references. Out of 43 only 10 were published within last 5 years.

Author Response

Dear Dr. Reviewer,

First of all, we are very grateful to you to for your suggestions to improve our manuscript. We are pleased to answer your questions and make revisions according to your suggestions.

Yes

Can be improved

Must be improved

Not applicable

Does the introduction provide sufficient background and include all relevant references?

( )

(x)

( )

( )

Is the research design appropriate?

(x)

( )

( )

( )

Are the methods adequately described?

( )

(x)

( )

( )

Are the results clearly presented?

( )

(x)

( )

( )

Are the conclusions supported by the results?

( )

( )

(x)

( )

Comments and Suggestions for Authors

The article presents an interesting study. However, I have some comments:

Title

I suggest changing the title to:

Canal Transportation and Volumetric Dentin Removal Abilities of Ni-Ti Rotary File Systems in Curved Primary Root Canals: CBCT Study

A: As authors we appreciate for your valuable comment. We agree with your remarkable comment and the title was changed as ‘’ Canal Transportation and Volumetric Dentin Removal Abilities of Ni-Ti Rotary File Systems in Curved Primary Root Canals: CBCT Study ‘’ in the revised manuscript.

Introduction

Please summarize first 10 lines into 1-2 sentences.

A: We would like to thank you for your constructive comment. Our changes regarding your comment is as indicated below.

The first 10 lines was summarized as  ‘‘ Particularly, in the cases of primary molars diagnosed with irreversible pulpitis or pulp necrosis, endodontic approaches (pulpectomy procedure) could be considered as an appropriate solution to provide maintenance of these teeth in the oral cavity until their required physiological shedding time in the pediatric dentistry field ’’ in the Introduction section.

Results

Please divide results into subsections.

A: We appreciate your valuable comment. According to your suggestion, the results section was divided into subsections consisting of canal transportation and volumetric dentin removal to make it more fluent and reader friendly.

Please explain in method section what do you mean with “Mean1’ and ‘Mean2’ (table 1).

A: We would like to thank you for your constructive comment. In method section we tried to explain the sequence between multiple comparisons. This explanation is indicated below with an example:

e.g: Comparison between Race Evo vs. R-Motion (Mean1, Mean2; respectively)

Mean1 presents the mean value of RaceEvo; Mean2 was presents the mean value of R-Motion.

Can you present table 1 as box and whiskers plot ?

A: Thank you for your valuable comment. Table 1 with all details is presented as box and whiskers plot in Figure 4.

Figure 4: Was there any statistically significant difference between PTG and manual group? The information is missing on the graph.

A: We appreciate for your supportive comment. There was no statistically difference between PTG and manual group. This information was added to the Figure 4 and represented in the text.

Please place capitation of the table above the table.

A: Thank you for this comment. Capitations of the both tables (Table 1 and Table 2 ) were placed above the tables.

Discussion

Lines 271-316 please divide this paragraph into smaller sections.

A: We would like to thank you for your constructive comment. According to your suggestion, lines 271-316 were divided into 4 smaller sections to make manuscript more reader friendly.

Conclusions

This conclusion No 3 do not answer the aim and is not supported by results.

A: We would like to thank you for your precious comment. The No3 conclusion statement  ‘’The CBCT could be used in curved root canals with its high diagnostic effectiveness and sensitivity in pediatric endodontics when the alternative methods is not adequate‘’ was removed from the conclusion section to make manuscript more fluent.

References

Please up-date references. Out of 43 only 10 were published within last 5 years.

A: Thank you for your valuable comment. Whole bibliography was checked and up-dated. Specifically, the references before 2010 year were tried to remove and changed with up-dated publications. The up-dated references were highlighted as red color in the Reference section.

Round 2

Reviewer 1 Report

The authors have implemented the required changes, substantially improving the deficiencies that the article presented in the initial version.

Author Response

Dear Dr. Reviewer,

As authors, we are very grateful to you to for your valuable comments and suggestions to improve our manuscript.

Open Review

English language and style

( ) Extensive editing of English language and style required
( ) Moderate English changes required
( ) English language and style are fine/minor spell check required
(x) I don't feel qualified to judge about the English language and style

Yes

Can be improved

Must be improved

Not applicable

Does the introduction provide sufficient background and include all relevant references?

(x)

( )

( )

( )

Is the research design appropriate?

(x)

( )

( )

( )

Are the methods adequately described?

(x)

( )

( )

( )

Are the results clearly presented?

(x)

( )

( )

( )

Are the conclusions supported by the results?

(x)

( )

( )

( )

Comments and Suggestions for Authors

The authors have implemented the required changes, substantially improving the deficiencies that the article presented in the initial version.

Reviewer 2 Report

Dear Authors,

Thank you for improving the manuscript. Please find below my comments.

Introduction

Please modify the first sentence after removal of previous sentences.

Results

Term “mean” in tables – please substitute it with the name of the file. Now it is confusing.

Figures – please remove the indicator ‘ns’.

Author Response

Dear Dr. Reviewer,

As authors, we are very grateful to you to for your valuable comments and suggestions to improve our manuscript. We are pleased to answer your questions and make revisions according to your suggestions.

Open Review

English language and style

( ) Extensive editing of English language and style required
( ) Moderate English changes required
( ) English language and style are fine/minor spell check required
(x) I don't feel qualified to judge about the English language and style

Yes

Can be improved

Must be improved

Not applicable

Does the introduction provide sufficient background and include all relevant references?

( )

(x)

( )

( )

Is the research design appropriate?

(x)

( )

( )

( )

Are the methods adequately described?

(x)

( )

( )

( )

Are the results clearly presented?

( )

(x)

( )

( )

Are the conclusions supported by the results?

(x)

( )

( )

( )

Comments and Suggestions for Authors

Dear Authors,

Thank you for improving the manuscript. Please find below my comments.

Introduction

Please modify the first sentence after removal of previous sentences.

A: We would like to thank you for your constructive comment. Our changes regarding your comment is as indicated below.

The first sentence was changed as  ‘‘Today, endodontic approaches (pulpectomy procedure) could be considered as an the most appropriate solution to provide maintenance of primary molars diagnosed with irreversible pulpitis or pulpal necrosis in the oral cavity until their required physiological shedding time ’’ to make a descriptive statement in the ‘Introduction’ section.

Results

Term “mean” in tables – please substitute it with the name of the file. Now it is confusing.

A: We would like to thank you for your precious comment. We have made changes in the Results section  according to the your comment and term “mean” in tables was substituted with the name of the file.

Figures – please remove the indicator ‘ns’.

A: We appreciate your valuable comment. All ‘ns’ indicators were removed from Figure 4 and Figure 6 and also figure legends in the manuscript.
